# An artificial intelligence approach for investigating multifactorial pain-related features of endometriosis

**Amber C. Kiser**[1], **Karen C. Schliep**[2], **Edgar Javier Hernandez**[1,3], **C. Matthew Peterson**[4], **Mark Yandell**[3], **Karen Eilbeck**[1]*

**1** Department of Biomedical Informatics, University of Utah, Salt Lake City, Utah, United States of America, **2** Department of Family and Preventative Medicine, University of Utah, Salt Lake City, Utah, United States of America, **3** Department of Human Genetics, Utah Center for Genetic Discovery, University of Utah, Salt Lake City, Utah, United States of America, **4** Department of Obstetrics and Gynecology, Division of Reproductive Endocrinology and Infertility, University of Utah, Salt Lake City, Utah, United States of America

* keilbeck@genetics.utah.edu

**Data Availability Statement:** The underlying data from this study cannot be shared publicly due to privacy restrictions as it contains protected health information. Data requests for the ENDO Study can

## Abstract

Endometriosis is a debilitating, chronic disease that is estimated to affect 11% of reproductive-age women. Diagnosis of endometriosis is difficult with diagnostic delays of up to 12 years reported. These delays can negatively impact health and quality of life. Vague, non-specific symptoms, like pain, with multiple differential diagnoses contribute to the difficulty of diagnosis. By investigating previously imprecise symptoms of pain, we sought to clarify distinct pain symptoms indicative of endometriosis, using an artificial intelligence-based approach. We used data from 473 women undergoing laparoscopy or laparotomy for a variety of surgical indications. Multiple anatomical pain locations were clustered based on the associations across samples to increase the power in the probability calculations. A Bayesian network was developed using pain-related features, subfertility, and diagnoses. Univariable and multivariable analyses were performed by querying the network for the relative risk of a postoperative diagnosis, given the presence of different symptoms. Performance and sensitivity analyses demonstrated the advantages of Bayesian network analysis over traditional statistical techniques. Clustering grouped the 155 anatomical sites of pain into 15 pain locations. After pruning, the final Bayesian network included 18 nodes. The presence of any pain-related feature increased the relative risk of endometriosis (p-value < 0.001). The constellation of chronic pelvic pain, subfertility, and dyspareunia resulted in the greatest increase in the relative risk of endometriosis. The performance and sensitivity analyses demonstrated that the Bayesian network could identify and analyze more significant associations with endometriosis than traditional statistical techniques. Pelvic pain, frequently associated with endometriosis, is a common and vague symptom. Our Bayesian network for the study of pain-related features of endometriosis revealed specific pain locations and pain types that potentially forecast the diagnosis of endometriosis.

be directed to Dr. Zhen Chen, email: chenzhe@mail.nih.gov, phone: 301-435-6934, from the NICHD/DIPHR Biospecimen Repository Access and Data Sharing (BRADS). The public GitHub repository amberkiser/endometriosis-pain-net contains the underlying code. The Bayesian network can be accessed via the web address: amber-kiser.shinyapps.io/ENDO-pain-app.

**Funding:** ACK is supported by training grant T15LM007124 from the National Library of Medicine. KCS is supported in part by the National Institute on Aging of the National Institutes of Health under Award Number K01AG058781. The ENDO (Endometriosis, Natural History, Diagnosis, and Outcomes) study was funded by the Intramural Research Program, Eunice Kennedy Shriver National Institute of Child Health and Human Development (NICHD), National Institutes of Health (contract numbers N01-DK-6-3428, N01-DK-6-3427, and 10001406-02). The funders had no role in study design, data collection and analysis, decision to publish, or preparation of the manuscript.

**Competing interests:** The authors have declared that no competing interests exist.

## Introduction

Endometriosis is a debilitating, chronic disease, estimated to affect 11% of reproductive-age women [1]. While it is primarily characterized by the presence of endometrial-like tissue outside the uterus, it is a complex disease, involving multiple anatomical systems and sites [2, 3]. Diagnosis of endometriosis is difficult, with delays of up to 12 years reported [2]. Vague, non-specific symptoms with multiple differential diagnoses contribute to this challenge. Misdiagnosis is common, with one study reporting 63% of affected women were told there was nothing wrong by at least one physician before ultimately receiving a diagnosis of endometriosis [4]. Additionally, the gold standard of diagnosis requires laparoscopic surgery—an invasive and potentially costly procedure [3, 5]. Long diagnostic delays can significantly impact the quality of life for affected women, leading to higher incidence of depression, reduced social activity, and increased healthcare costs [5]. Thus, earlier diagnosis and treatment are essential to improving the quality of life for these patients, especially by minimizing pain and treating infertility.

Pain is potentially an indicator of multiple health problems, not just endometriosis. Pelvic pain affects 80–98% of women with endometriosis [4, 6]; however, this common pain type manifests in several other disorders, including uterine fibroids, ovarian cysts, pelvic adhesions, and infectious diseases [7–12]. In some cases, chronic pelvic pain cannot be attributed to any clinical pathology, even after diagnostic laparoscopy [13]. Women with endometriosis often experience multiple types of pain, including dysmenorrhea (painful menstrual cramps), dyspareunia (pain with intercourse), chronic pelvic pain, dysuria (pain with urination), and dyschezia (pain with bowel movements) [14, 15]. Endometriosis-associated pain manifests in complex types, patterns, and intensities.

Identifying additional clinical indications of endometriosis holds promise for facilitating a shift from surgical diagnosis to less-invasive methods, decreasing the diagnostic delay without increasing the incidence of misdiagnosis [3]. Some previous studies examining pain in endometriosis have focused on types of pain [2, 16, 17]; in contrast, others have investigated possible relationships between anatomical locations of pain and the location of endometrial lesions although results were inconclusive [18, 19]. The dataset used in this study was previously analyzed with traditional statistical analysis techniques. The data contained meaningful associations; however, the granularity of the 155 anatomical pain sites made it difficult to interpret these preliminary results. Thus, further investigation with more advanced data science methods was warranted.

The objective for this study was to apply advanced data mining and artificial intelligence (AI) methods to delve deeper into the relationships between pain-related features, subfertility, and gynecological diagnoses, including endometriosis. We sought to demonstrate the capability of AI to perform analyses and discover associations that are more difficult to conduct using traditional statistical methods. This preliminary model was created to show the potential for AI to help expedite endometriosis diagnosis, using clinical factors in unsupervised clustering and Bayesian networks.

## Materials and methods

### Data collection

Data were previously collected from January 24, 2007, until August 7, 2009, as part of the ENDO (Endometriosis, Natural History, Diagnosis, and Outcomes) Study conducted in Salt Lake City, Utah, and San Francisco, California and described previously [1]. Institutional review board approval (Institutional Review Board, University of Utah) and written informed consent from all participants was obtained. The operative cohort included 473 women undergoing laparoscopy or laparotomy for various surgical indications, including pelvic pain (44%),

pelvic mass (16%), irregular menses (13%), fibroids (10%), tubal ligation (10%) and infertility (7%). To prevent recall bias, women who had a prior surgical diagnosis of endometriosis were excluded. Women were also excluded if they were: pregnant, currently breastfeeding for 6 months or more, given injectable hormones within the past 2 years, or diagnosed with cancer other than non-melanoma skin cancer.

## Gynecological pathology assessment

Surgeons completed a standardized operative report immediately after surgery to capture gynecologic and pelvic pathology. Endometriosis was diagnosed using the clinical gold standard of surgically visualized disease [5, 20, 21] while its severity was based upon the revised American Society for Reproductive Medicine's (rASRM) staging criteria [20]. We used the standardized operative form to assess endometriosis typology: superficial endometriosis, ovarian endometriomas, and deep infiltrating endometriosis [22].

## Pain evaluation

Prior to surgery, participants answered in-depth survey questions regarding their pain experiences. Women were asked whether they experienced pain lasting greater than 6 months that was either cyclic (i.e., painful menstrual cramps not relieved by over-the-counter medications) or chronic (i.e., pain located in or near the bladder or vaginal canal not associated with menses). They rated different sources of pain on an 11-point visual analog scale (VAS) with 0 denoting no pain to 10 denoting the most severe pain imaginable using a standardized questionnaire [23]. To evaluate subfertility, participants indicated if they had ever tried to get pregnant for more than 6 months or sought treatment for infertility. We discretized the responses to create the clinical features represented in S1 Table. We determined the most appropriate pain threshold to dichotomize the data by identifying the pain threshold that maximized the specificity of the Bayesian network. Participants answered all questions, resulting in no missing data. Additionally, prior to surgery, participants indicated on a computerized anatomical map where they felt pain regularly. The VAS and anatomical map were adapted from the Pelvic Pain Assessment Form created by the International Pelvic Pain Society [24, 25].

## Clustering pain map sites

The anatomical pain map comprised 155 total sites (60 front / 60 back / 35 perineal). We performed neighbor-joining (NJ) clustering to aggregate similar pain sites [26]. The NJ algorithm is an agglomerative, or bottom-up, clustering method that groups similar data points based on a given distance metric. We used the Jaccard distance metric to quantify similarities between sites [27]. We employed the commonly used "elbow" method to determine the appropriate number of clusters by plotting the number of clusters against a given threshold (the maximum distance separating the sites within each cluster) [28]. To perform NJ clustering, we used the R package DECIPHER, version 2.22.0 [29]. The neighbor-joining algorithm produced an unrooted, non-ultrametric tree. We used the threshold determined by the elbow method to create the clusters. To reduce extraneous variables, grouped anatomical locations where fewer than 5% of participants (N<25) indicated they felt pain were removed from further analysis.

## Bayesian network

Bayesian networks, a form of AI, are directed graphs that model conditional dependencies between variables [30, 31]. Each node in the network represents a variable with an attached conditional probability table, while each edge in the network represents a conditional dependency

between nodes. Using these conditional dependencies and Bayesian inference, a query of the network can return the probability of an event conditioned upon the presence or absence of other variables in the network. Bayesian networks can model multiple dependencies between variables, account for non-additive risk propagation, and serve as a form of explainable AI [32].

We developed a Bayesian network using diagnoses and symptoms, including pain types (generated from the survey questions), pain locations (generated from the clustering), and subfertility. The structure of the network was learned from the data, using a hill-climbing algorithm, which iteratively improved the Bayesian Information Criterion (BIC) score until a local maxima was reached [33]. The conditional probability tables were learned from the data using Bayesian posterior estimation. We developed the network using the R package, bnlearn, version 4.7 [34]. For visualization as an undirected graph, the network was moralized. To moralize a network, an undirected edge was added between the parents of each node, after which all directional arrows were dropped [35].

## Performance analysis

We evaluated the performance of the network to predict diagnoses using sensitivity, specificity, and precision. The network was bootstrapped, and out-of-bag samples were used to calculate the performance metrics. We calculated the mean and 95% confidence intervals for the performance metrics after 1000 iterations of bootstrapping.

We evaluated the relative risk (RR) of a postoperative diagnosis, given the presence of a single symptom by querying the Bayesian network. To demonstrate the potential of the Bayesian network to model relationships between variables, we queried the network for the RR of a postoperative diagnosis, given multiple symptoms. The network was bootstrapped for 1000 iterations to obtain the mean and 95% confidence intervals for each RR. We calculated the RR using the R package, gRain, version 1.3.6 [35].

## Sensitivity analyses

To compare the Bayesian network analysis with traditional statistical analysis, we evaluated the significance of pain-related features as they related to the postoperative diagnoses using Fisher's exact test. We adjusted the p-values to control for the false discovery rate. We used Boschloo's exact test to evaluate pairwise comparisons for any significant associations [36].

To further demonstrate the advantages of the Bayesian network, we performed an inverted analysis. We queried the net for the RR of subfertility given different diagnoses, defined as the risk of subfertility when a diagnosis is present compared to the risk of subfertility when no pathology is present.

We evaluated the ability of the Bayesian network to discriminate between rASRM stages, categorized as early-stage endometriosis (rASRM stages I and II) or late-stage endometriosis (rASRM stages III and IV), as well as endometriosis typology, categorized as superficial or deep infiltrating (DIE) and endometriomas [22]. We queried the net for the RR of a postoperative diagnosis as previously described. We used the independent samples t-test to evaluate if the RR between each stage or typology were significantly different. We adjusted the p-values to control for the false discovery rate.

## Results

### Cohort description

Postoperative diagnoses included: endometriosis (n = 190; 40.2%), normal pelvis (n = 122; 25.8%), uterine fibroids (n = 59; 12.5%), benign ovarian cysts (n = 52; 11.0%), and other

gynecological pathologies (n = 50; 10.6% [pelvic adhesions (n = 37), congenital Müllerian anomalies (n = 10) and neoplasm (n = 3)]) [1]. S1 Fig illustrates the distribution of VAS pain scores for each pain type.

## Clustering

Collecting the pain data at a very granular level and then clustering pain locations based on participant responses enabled the grouping of pain locations. The elbow method determined 15 clusters with a threshold of 0.6375 as the maximum distance separating the sites within each cluster was optimal for this data (S2 Fig). A polar dendrogram visualizes the clustering in S3 Fig. This data-driven approach generated clusters that nonuniformly divided the anatomical landscape. The clusters could be categorized in several ways, including singular (cluster 1— front of neck), discontinuous (cluster 12—front of both arms), encapsulated (cluster 2—urethra), asymmetrical (cluster 8—sternum and arm) or anteroposterior (clusters 3 and 4—abdomen and back) (see Fig 1).

The underlying anatomy could explain some of the apparent symmetry and asymmetry. For example, while cluster 7 (breasts) was symmetrical, clusters 6 and 9 (right and left epigastrium) were asymmetrical. This could be explained by the underlying anatomy of the abdomen, as the liver and gallbladder lie on the right side and the head of the pancreas and spleen on the left side. Clusters 5 (right hypochondrium and arm) and 8 (sternum and arm) were both non-contiguous and asymmetrical. Cluster 14 encompassed multiple non-contiguous sites, including the lower limbs, inner thighs, upper back, and waist. The elbow chart indicated this cluster included the least informative sites, as splitting this cluster would not result in much information gain. S2 Table describes the clusters.

## Bayesian network

The final Bayesian network, represented in Fig 2, included 18 nodes after pruning noninformative variables. For endometriosis diagnosis, the sensitivity of the network was 0.814 [95% CI 0.807, 0.821], the specificity was 0.421 [95% CI 0.413, 0.430], and the precision was 0.490

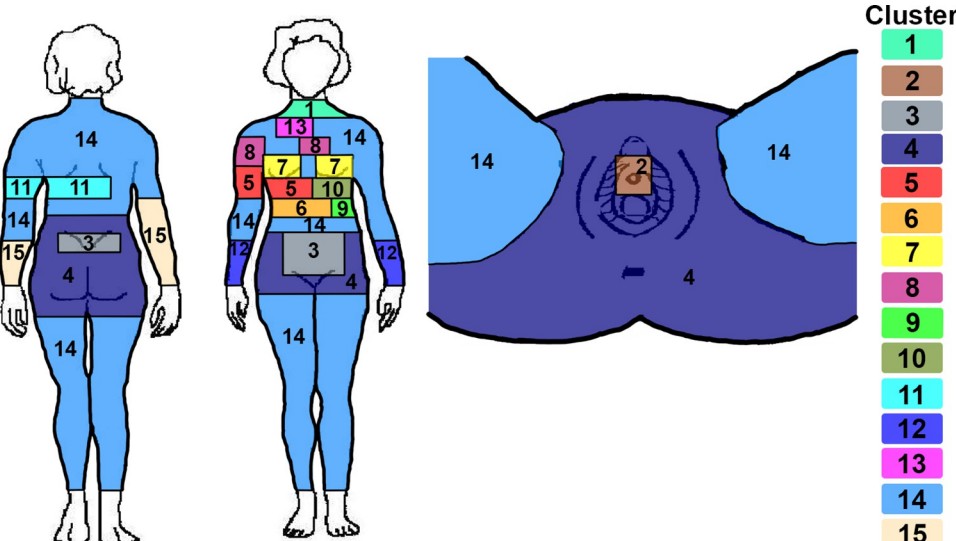

**Fig 1. Anatomical map of pain location clusters.** Colors on the anatomical map denote clustered locations of self-reported pain. The anatomical map used in the ENDO study was modified from the International Pelvic Pain Society.

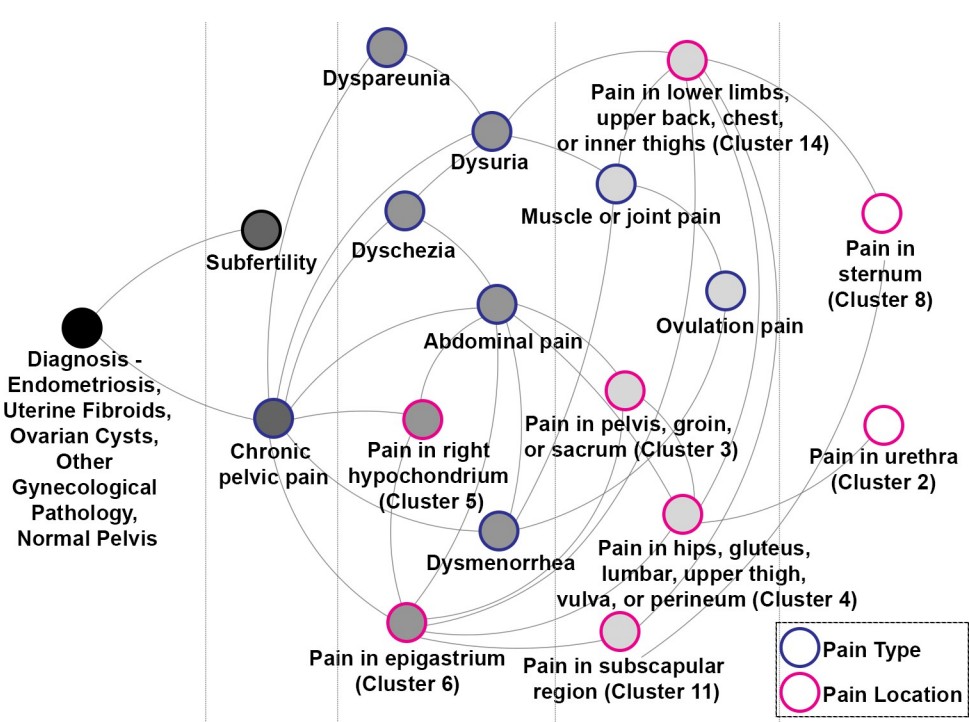

**Fig 2. Bayesian network for pain-related features of endometriosis.** Nodes correspond to pain-related features, subfertility, and diagnosis while the lines represent conditional dependencies between the features. Pain types are outlined in blue, while pain locations are outlined in pink. The color of the node indicates the proximity to the diagnosis node, including 1st, 2nd, 3rd, and 4th degree connections. The net has been moralized.

[95% CI 0.487, 0.493]. In the univariable analysis, the presence of any symptom, including chronic pelvic pain, subfertility, and dyspareunia, increased the RR of endometriosis (p-value < 0.001) (Table 1).

As shown in Table 2, the combination of multiple symptoms increased the RR of endometriosis while decreasing the RR of other diagnoses. However, the absence of subfertility decreased the RR of endometriosis and increased the RR of other diagnoses, most notably benign ovarian cysts.

A web application to explore and query the Bayesian network is deployed to a public server: amber-kiser.shinyapps.io/ENDO-pain-app. We developed and deployed the application using the R package shiny, version 1.7.1 [37]. The S7–S10 Figs contain screenshots and instructions for how to access the application.

## Sensitivity analyses

The first sensitivity analysis allowed us to compare the performance of our Bayesian network against a more traditional statistical method, Fisher's exact test, used in previous studies of endometriosis [23]. Fisher's exact test identified nine clinical features significantly associated with endometriosis, including dysmenorrhea, dyspareunia, chronic pelvic pain, subfertility, ovulation pain, pain in epigastrium (Cluster 6), dyschezia, pain in sternum (Cluster 8), and pain in subscapular region (Cluster 11). S3 Table provides the full results from Fisher's exact test and Boschloo's exact test for pairwise comparisons.

In the second sensitivity analysis, we performed an inverted analysis, where the diagnoses and symptoms were reversed, to further demonstrate the advantages of using a Bayesian network. We found that the RR of subfertility was highest with a diagnosis of endometriosis. The

**Table 1. Relative risk of a diagnosis given a pain type or location.**

| Symptom | Endometriosis | Uterine Fibroids | Benign Ovarian Cysts | Other Gynecological Pathology | Normal Pelvis |
|---|---|---|---|---|---|
| Chronic pelvic pain | 2.047 (2.015, 2.079) | 0.527 (0.514, 0.540) | 1.062 (1.043, 1.080) | 0.797 (0.784, 0.810) | 0.653 (0.643, 0.664) |
| Subfertility | 1.493 (1.467, 1.518) | 0.788 (0.774, 0.801) | 0.596 (0.574, 0.618) | 0.917 (0.903, 0.930) | 0.839 (0.829, 0.849) |
| Dyspareunia | 1.273 (1.265, 1.281) | 0.753 (0.745, 0.760) | 1.004 (0.997, 1.012) | 0.899 (0.892, 0.906) | 0.830 (0.825, 0.836) |
| Dysmenorrhea | 1.260 (1.249, 1.271) | 0.797 (0.789, 0.804) | 0.985 (0.976, 0.994) | 0.888 (0.879, 0.897) | 0.854 (0.850, 0.859) |
| Abdominal pain for at least 12 weeks | 1.243 (1.236, 1.250) | 0.785 (0.778, 0.791) | 1.006 (0.999, 1.013) | 0.900 (0.894, 0.906) | 0.860 (0.855, 0.865) |
| Pain in epigastrium (Cluster 6) | 1.173 (1.166, 1.180) | 0.775 (0.763, 0.788) | 1.034 (1.023, 1.044) | 0.886 (0.875, 0.897) | 0.919 (0.910, 0.929) |
| Dysuria | 1.173 (1.167, 1.178) | 0.822 (0.816, 0.828) | 1.008 (1.003, 1.013) | 0.923 (0.919, 0.928) | 0.882 (0.878, 0.887) |
| Dyschezia | 1.130 (1.124, 1.135) | 0.871 (0.865, 0.876) | 0.999 (0.994, 1.003) | 0.941 (0.936, 0.946) | 0.909 (0.904, 0.913) |
| Ovulation pain | 1.127 (1.120, 1.134) | 0.883 (0.879, 0.887) | 0.991 (0.986, 0.996) | 0.937 (0.932, 0.942) | 0.922 (0.919, 0.926) |
| Pain in subscapular region (Cluster 11) | 1.115 (1.111, 1.119) | 0.841 (0.832, 0.850) | 1.019 (1.012, 1.025) | 0.917 (0.909, 0.925) | 0.944 (0.937, 0.950) |
| Pain in right hypochondrium (Cluster 5) | 1.098 (1.094, 1.103) | 0.851 (0.841, 0.860) | 1.032 (1.022, 1.041) | 0.921 (0.913, 0.929) | 0.951 (0.944, 0.957) |
| Pain in sternum (Cluster 8) | 1.083 (1.075, 1.091) | 0.881 (0.872, 0.891) | 1.016 (1.008, 1.023) | 0.930 (0.921, 0.940) | 0.957 (0.952, 0.963) |
| Pain in lower limbs, upper back, chest, or inner thighs (Cluster 14) | 1.074 (1.072, 1.077) | 0.909 (0.905, 0.913) | 1.009 (1.006, 1.013) | 0.954 (0.950, 0.957) | 0.959 (0.956, 0.962) |
| Pain in hips, gluteus, lumbar, upper thigh, vulva, or perineum (Cluster 4) | 1.074 (1.070, 1.077) | 0.930 (0.926, 0.934) | 0.997 (0.994, 0.999) | 0.969 (0.966, 0.972) | 0.948 (0.946, 0.951) |
| Muscle or joint pain | 1.067 (1.064, 1.070) | 0.927 (0.924, 0.930) | 1.001 (0.998, 1.004) | 0.965 (0.963, 0.968) | 0.954 (0.951, 0.956) |
| Pain in urethra (Cluster 2) | 1.040 (1.036, 1.043) | 0.960 (0.952, 0.969) | 0.994 (0.991, 0.997) | 0.979 (0.976, 0.982) | 0.971 (0.968, 0.973) |
| Pain in pelvis, groin, and sacrum (Cluster 3) | 1.030 (1.027, 1.033) | 0.997 (0.989, 1.004) | 0.990 (0.987, 0.993) | 1.004 (0.999, 1.009) | 0.968 (0.966, 0.971) |
| Baseline | 1.000 (1.000, 1.000) | 1.000 (1.000, 1.000) | 1.000 (1.000, 1.000) | 1.000 (1.000, 1.000) | 1.000 (1.000, 1.000) |

This table demonstrates univariable analysis with the Bayesian network. The numbers represent mean estimates of the relative risk (RR) of the specified diagnosis, i.e., endometriosis, uterine fibroids, benign ovarian cysts, other gynecological pathology, or a normal pelvis, given the presence of a single symptom with 95% confidence intervals shown in parentheses. Darker red denotes increased RR, while the darker blue denotes decreased RR. For the definition of RR, see S4 Fig.

RR of subfertility was most decreased when a normal pelvis was present; however, having a diagnosis of benign ovarian cysts or uterine fibroids decreased the RR of subfertility. The results are illustrated in the forest plot in S6 Fig.

In our final sensitivity analysis, we evaluated the ability of the net to discriminate between rASRM stage and endometriosis typology. Results are presented in S4 and S5 Tables. The RRs for early-stage endometriosis were statistically different from the RRs for late-stage endometriosis in 8 out of 17 cases. The RRs for superficial endometriosis were statistically different from the RR for DIE or endometriomas in 16 out of 17 cases. This demonstrates the potential for discrimination between stages or typology. However, more data from each category would be required to develop a network for this purpose.

**Table 2. Relative risk of a diagnosis given multiple pain types or locations.**

| Prior Condition(s) | Endometriosis | Uterine Fibroids | Benign Ovarian Cysts | Other Gynecological Pathology | Normal Pelvis |
|---|---|---|---|---|---|
| Pain in epigastrium | 1.176 (1.169, 1.182) | 0.765 (0.752, 0.779) | 1.043 (1.031, 1.055) | 0.877 (0.865, 0.889) | 0.926 (0.916, 0.936) |
| Pain in epigastrium + Dyspareunia | 1.420 (1.409, 1.430) | 0.613 (0.600, 0.625) | 1.047 (1.033, 1.061) | 0.814 (0.802, 0.827) | 0.801 (0.789, 0.812) |
| Pain in epigastrium + Dyspareunia + Subfertility | 2.050 (2.013, 2.086) | 0.488 (0.475, 0.502) | 0.654 (0.629, 0.678) | 0.726 (0.710, 0.743) | 0.673 (0.661, 0.686) |
| Pain in epigastrium + Dyspareunia + No Subfertility | 1.051 (1.030, 1.072) | 0.837 (0.813, 0.861) | 3.139 (2.281, 3.996) | 0.948 (0.925, 0.972) | 0.973 (0.954, 0.993) |

This table demonstrates multivariable analysis with the Bayesian network. The numbers represent mean estimates of the relative risk (RR) of the specified diagnosis, i.e., endometriosis, uterine fibroids, benign ovarian cysts, other gynecological pathology, or a normal pelvis, given a combination of symptoms (presence or absence of a single or multiple symptoms) with 95% confidence intervals shown in parentheses. Darker red denotes increased RR, while the darker blue denotes decreased RR. For the definition of RR, see S5 Fig.

## Discussion

We used a combined approach of clustering and Bayesian networks to demonstrate the power of AI to discover associations between pain-related features, subfertility, and endometriosis. The network identified the transitive relationships between postoperative diagnoses and pain-related features (Tables 1 and 2). The presence of any pain increased the RR of endometriosis compared with benign ovarian cysts, uterine fibroids, other gynecological pathology, or a normal pelvis.

To date, several studies have focused on the relationship between pain and diagnosis, but relatively few have tried to understand how specific anatomical locations and types of pain combine to forecast endometriosis. The few studies incorporating anatomical locations of pain have usually attempted to relate pain to endometrial lesion location with inconclusive results [18, 19]. In 2010, Ballard et al. conducted a study of women with chronic pelvic pain to evaluate how specific symptoms and descriptions of pain can facilitate endometriosis diagnosis [38]. They included an anatomical pain map and grouped 100 sites into 15 areas, although no description was provided of how this grouping was determined. None of the anatomical areas of pain were significantly associated with endometriosis in their study, as women with endometriosis and women with a normal pelvis felt pain indiscriminately.

In our study, we saw a similar trend for some pain locations. While a higher number of women with endometriosis reported pain in the lower abdominal and pelvic region (61.6%), half of the women with a normal pelvis also reported pain in these regions. Despite this, with our Bayesian network analysis, we demonstrated that pain in certain anatomical locations can significantly increase the RR of endometriosis. We found that the combination of specific anatomical locations of pain with specific types of pain enables greater diagnostic discrimination.

The Bayesian network identified pain in the epigastrium, subscapular region, sternum, and right hypochondrium as significant for an endometriosis diagnosis. While previous literature has not associated these pain locations with pelvic endometriosis, pain in comparable locations has been reported in thoracic or diaphragmatic endometriosis [39–42]. Piccus et al. found that diaphragmatic endometriosis patients reported pain in the chest, upper abdomen, shoulder, and upper back [43]. Other reports describe diaphragmatic endometriosis in patients with right chest, shoulder, and hypochondrium pain [44, 45]. Pain in the epigastrium, subscapular region, sternum, and right hypochondrium could accelerate the diagnosis by indicating potential extrapelvic endometriosis; however, this finding will need to be validated in future research.

The symptoms of endometriosis, including dyschezia and abdominal pain, are also common to irritable bowel syndrome (IBS) and constipation. Several studies have shown an association between endometriosis and IBS [14, 46]. However, it is not known if this association is due to misdiagnosis or a true comorbidity. Dyschezia was an important pain type in our study as it was associated with an increased RR of endometriosis and significantly associated with endometriosis in the sensitivity analysis. Future studies can potentially differentiate endometriosis from IBS or other conditions like fibromyalgia, interstitial cystitis, and myofascial pain.

In the first sensitivity analysis, we compared the features significantly associated with endometriosis using Fisher's exact test and the Bayesian network. When comparing these results, the nine features identified as significant in Fisher's exact test increased the RR of endometriosis in the Bayesian network. However, we also identified other features, including abdominal pain, dysuria, and pain in the right hypochondrium that increased the RR of endometriosis when included in the Bayesian network, but were not significant in Fisher's exact test. These results demonstrate the ability of the net to uncover previously undetected relationships that would be considered insignificant in our data using traditional statistical methods.

Further evidence of the benefits of an AI-based analysis was demonstrated in the multivariable analysis (Table 2). Combining multiple pain-related features and subfertility added discrimination power, allowing the Bayesian network to differentiate between endometriosis and other diagnoses. This multivariable analysis would be more difficult to perform using traditional statistical methods as it is much harder to consider the combined contributions of several variables, especially when those are not necessarily additive.

Prior uses of graphical models and Bayesian networks in endometriosis research include investigations of risk factors and treatment efficacy [47–49]. However, we are aware of no other studies that have implemented Bayesian networks to evaluate pain-related features of endometriosis, as most have used logistic regression to investigate clinical features of endometriosis. Unlike regression-based models, Bayesian networks can be used for univariable or multivariable analysis while considering the relationships between variables, and importantly they are explainable and handle missing data [32].

We acknowledge that our study has limitations. While we tried avoiding bias in the study population, we only included women who were undergoing laparoscopy or laparotomy for a surgical indication, potentially excluding some asymptomatic women. The study population was predominantly white, well-educated women with access to healthcare. In future studies, we could use additional, external, and more diverse data to validate these results. External validation using large, heterogeneous data will be vital prior to clinical implementation. Future studies could explore factors that mediate pain experiences, including pain medications and psychological conditions, like anxiety and depression. Pain duration as well as pain description (i.e., sharp, dull, radiating) are potentially important variables to include in future studies along with family history of endometriosis and response to previous hormonal treatment. With additional data, we could investigate comorbidities that may alter pain presentation in endometriosis, including irritable bowel syndrome and constipation.

## Conclusions

Clustering anatomical pain sites and developing a Bayesian network, a form of AI, to study pain-related features of endometriosis demonstrated that combining specific pain locations with pain types may be useful for the diagnosis of endometriosis. We demonstrated that differing combinations of pain-related features resulted in differing RRs among women with endometriosis, benign ovarian cysts, uterine fibroids, other gynecological pathology, and a normal pelvis. These results suggest the use of AI can provide novel insights. We built a prototype web

application to demonstrate the potential of the Bayesian network to assist in diagnosis after external validation. A recent qualitative study of endometriosis-related pain suggested that looking beyond cyclical pain to other pain experiences may lead to faster diagnosis and more effective treatment [15]. Our Bayesian network, incorporating a variety of pain-related features of endometriosis, advances this goal.

## Supporting information

**S1 Table. Clinical features with corresponding survey questions.** These questions were used to determine the presence of specific clinical features in participants of the ENDO study. (Buck Louis GM, Hediger ML, Peterson CM, et al. Incidence of endometriosis by study population and diagnostic method: the ENDO study. Fertil Steril. 2011;96(2):360–365). (PDF)

**S2 Table. Clustered pain map areas.** Clusters were grouped as described in the methods. Location lists the anatomical sites included in each cluster. Asterisks in the Custer ID column denote clusters with N<25 participants. (PDF)

**S3 Table. Characteristics of the study sample by postoperative diagnosis.** This table presents a univariate analysis done with traditional statistical techniques. Significant differences were assessed using Fisher's Exact Test. P-value adjusted for the false discovery rate. Pairwise comparisons for any significant associations were evaluated with Boschloo's Exact Test. [a,b,c,d,e,f]Significantly different pairwise values p-value < 0.05. (PDF)

**S4 Table. Relative risk of an endometriosis rASRM stage, given a symptom.** For the definition of relative risk (RR), see S4 Fig. Mean estimates with 95% confidence intervals are shown in parentheses. Significant differences were assessed using the independent samples t-test. P-value adjusted for the false discovery rate. * p-value < 0.05; *** p-value < 0.001. (PDF)

**S5 Table. Relative risk of an endometriosis typology, given a symptom.** For the definition of relative risk (RR), see S4 Fig. Mean estimates with 95% confidence intervals are shown in parentheses. Significant differences were assessed using the independent samples t-test. P-value adjusted for the false discovery rate. * p-value < 0.05; *** p-value < 0.001. (PDF)

**S1 Fig. Distribution of VAS pain scores.** The distribution of VAS pain scores for each pain type. We determined a pain threshold of 1 was most appropriate for dichotomization of pain scores (see S1 Table). (TIF)

**S2 Fig. The elbow method identifies 15 clusters as optimal.** The elbow method requires identifying the point in the chart where a sharp bend, or "elbow," occurs. This is determined to be the optimal number of clusters that are informative. The threshold represents the maximum distance separating sites within each cluster. In this chart, an elbow can be identified at 15 clusters, corresponding to a threshold of 0.6375. (TIF)

**S3 Fig. Polar dendrogram of anatomical sites used to produce 15 pain locations.** The neighbor-joining algorithm produced an unrooted tree. For visualization as a dendrogram, this tree

has been rooted at the midpoint.
(TIF)

**S4 Fig. Univariate relative risk.** (A) When evaluating the relative risk given a single pain-related feature, relative risk is defined as the absolute risk of having a diagnosis given the presence of a pain type or location compared to the absolute risk of having a diagnosis given the absence of a pain type or location. (B) For example, the relative risk of endometriosis when dyspareunia is present is calculated from the absolute risk of endometriosis when dyspareunia is present divided by the absolute risk of endometriosis when dyspareunia is absent.
(TIF)

**S5 Fig. Multivariate relative risk.** (A) When evaluating the relative risk in the presence or absence of multiple pain-related features, relative risk is defined as the absolute risk of having a diagnosis given the presence or absence of a pain type or location compared to the absolute risk of having a diagnosis given the opposite condition(s). (B) For example, the relative risk of endometriosis when pain in the epigastrium and dyspareunia are present and subfertility is absent is calculated from the absolute risk of endometriosis when pain in the epigastrium and dyspareunia are present and subfertility is absent divided by the absolute risk of endometriosis when pain in the epigastrium and dyspareunia are absent and subfertility is present.
(TIF)

**S6 Fig. Differential contribution of key exposures to the relative risk of subfertility.** Relative risk of subfertility given a specific diagnosis. This forest plot illustrates the relative risk of subfertility, defined as the absolute risk of subfertility when a specific diagnosis is present compared to the absolute risk of subfertility when a specific diagnosis is absent. Error bars represent the 95% confidence intervals. Values to the right of the dotted line indicate an increased relative risk, whereas values to the left indicate a decreased relative risk.
(TIF)

**S7 Fig. Landing page of the Bayes Net Explorer app.** The Bayes Net Explorer app visualizes the calculated Bayesian network and allows users to run any combination of queries, accessible at amber-kiser.shinyapps.io/ENDO-pain-app.
(TIF)

**S8 Fig. A query performed using the Bayes Net Explorer.** The probability and relative risk of postoperative diagnoses are returned, given the presence of ovulation pain.
(TIF)

**S9 Fig. A multimorbid query performed using the Bayes Net Explorer.** The probability and relative risk of postoperative diagnoses are returned, given the presence of ovulation pain and absence of subfertility.
(TIF)

**S10 Fig. An inverted query performed using the Bayes Net Explorer.** The probability and relative risk of subfertility is returned, given the presence of an endometriosis diagnosis, dyspareunia, and chronic pelvic pain.
(TIF)

## Acknowledgments

The authors would like to thank Evan Christensen for his assistance in proofreading and editing the manuscript.

## Author Contributions

**Conceptualization:** Amber C. Kiser, Karen C. Schliep, C. Matthew Peterson, Mark Yandell, Karen Eilbeck.

**Data curation:** Amber C. Kiser, Karen C. Schliep.

**Formal analysis:** Amber C. Kiser, Edgar Javier Hernandez.

**Investigation:** C. Matthew Peterson.

**Methodology:** Mark Yandell.

**Software:** Amber C. Kiser, Edgar Javier Hernandez.

**Supervision:** Mark Yandell, Karen Eilbeck.

**Visualization:** Amber C. Kiser.

**Writing – original draft:** Amber C. Kiser, Mark Yandell, Karen Eilbeck.

**Writing – review & editing:** Amber C. Kiser, Karen C. Schliep, Edgar Javier Hernandez, C. Matthew Peterson, Mark Yandell, Karen Eilbeck.

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
