## [Decision Letter · Decision Letter 0]

15 Nov 2023

PONE-D-23-25577An Artificial Intelligence Approach for Investigating Multifactorial Pain-Related Features of EndometriosisPLOS ONE

Dear Dr. Eilbeck,

Thank you for submitting your manuscript to PLOS ONE. After careful consideration, we feel that it has merit but does not fully meet PLOS ONE’s publication criteria as it currently stands. Therefore, we invite you to submit a revised version of the manuscript that addresses the points raised during the review process.

Please submit your revised manuscript by Dec 30 2023 11:59PM If you will need more time than this to complete your revisions, please reply to this message or contact the journal office at plosone@plos.org. Please include the following items when submitting your revised manuscript:A rebuttal letter that responds to each point raised by the academic editor and reviewer(s). You should upload this letter as a separate file labeled 'Response to Reviewers'.A marked-up copy of your manuscript that highlights changes made to the original version. You should upload this as a separate file labeled 'Revised Manuscript with Track Changes'.An unmarked version of your revised paper without tracked changes. You should upload this as a separate file labeled 'Manuscript'.If applicable, we recommend that you deposit your laboratory protocols in protocols.io to enhance the reproducibility of your results. Protocols.io assigns your protocol its own identifier (DOI) so that it can be cited independently in the future. For instructions see: https://journals.plos.org/plosone/s/submission-guidelines#loc-laboratory-protocols. Additionally, PLOS ONE offers an option for publishing peer-reviewed Lab Protocol articles, which describe protocols hosted on protocols.io. Read more information on sharing protocols at https://plos.org/protocols?utm_medium=editorial-email&utm_source=authorletters&utm_campaign=protocols.

We look forward to receiving your revised manuscript.

Kind regards,

Federico Romano, M.D., Ph.D.

Academic Editor

PLOS ONE

Journal Requirements:

Did you know that depositing data in a repository is associated with up to a 25% citation advantage (https://doi.org/10.1371/journal.pone.0230416)? If you’ve not already done so, consider depositing your raw data in a repository to ensure your work is read, appreciated and cited by the largest possible audience. You’ll also earn an Accessible Data icon on your published paper if you deposit your data in any participating repository (https://plos.org/open-science/open-data/#accessible-data).

"The authors would like to thank Evan Christensen for his assistance in proofreading and editing the manuscript. We want to acknowledge support from the National Library of Medicine training grant (T15LM007124). We thank the Pedigree and Population Resource of Huntsman Cancer Institute, University of Utah (funded in part by the Huntsman Cancer Foundation) for its role in the ongoing collection, maintenance and support of the Utah Population Database (UPDB). We also acknowledge partial support for the UPDB through grant P30 CA2014 from the National Cancer Institute, University of Utah and from the University of Utah’s program in Personalized Health and Center for Clinical and Translational Science."

"ACK is supported by training grant T15LM007124 from the National Library of Medicine. KCS is supported in part by the National Institute on Aging of the National Institutes of Health under Award Number K01AG058781. The ENDO (Endometriosis, Natural History, Diagnosis, and Outcomes) study was funded by the Intramural Research Program, Eunice Kennedy Shriver National Institute of Child Health and Human Development (NICHD), National Institutes of Health (contract numbers N01-DK-6-3428, N01-DK-6-3427, and 10001406-02). The funders had no role in study design, data collection and analysis, decision to publish, or preparation of the manuscript."

Reviewers' comments:

Reviewer's Responses to Questions

**Comments to the Author**

1. Is the manuscript technically sound, and do the data support the conclusions?

Reviewer #1: Yes

Reviewer #2: Yes

Reviewer #3: Yes

2. Has the statistical analysis been performed appropriately and rigorously? 

Reviewer #1: I Don't Know

Reviewer #2: Yes

Reviewer #3: Yes

3. Have the authors made all data underlying the findings in their manuscript fully available?

Reviewer #1: Yes

Reviewer #2: Yes

Reviewer #3: Yes

4. Is the manuscript presented in an intelligible fashion and written in standard English?

Reviewer #1: Yes

Reviewer #2: Yes

Reviewer #3: Yes

5. Review Comments to the Author

Reviewer #1: Dear Author

1-As we know some endometriosis patients do not complain pain and just have fertility problems, how can we use this approach for diagnosis ?

2-In figure 2 some clusters such as number 1,7,12,...were not included.

3-why do you consider region neck,breast,arm,sternum as pain region?These area are not site of endometriosis pain

4-What do you think about including duration of pain,response to previous hormonal treatment,family history of endometriosis to your approach?

Reviewer #2: Dear Editor,

Thanks for selecting me as a manuscript referee entitled " An Artificial Intelligence Approach for Investigating Multifactorial Pain-Related Features of Endometriosis". according to the following comments, the manuscript can be re -reviewed on the condition of modifying the following.

1. The numbers in the tables should be specified, for example which RR number is.

2. Explain the tables to make what they predict.

3. In the whole article, there is a lot of disruption and the clinical application of artificial intelligence in the treatment and diagnosis of endometriosis is not clear.

Reviewer #3: In Data Collection:

1-In the data collection, it was said that the inclusion criteria for the study were patients who underwent laparotomy or laparoscopy for example TL, myomectomy etc., Was the pain questionnaire provided to the patients "before" all of the surgery?

2-Why was it necessary to exclude patients with endometriosis from the study?

3- Is it possible to perform an analysis based on the relationship between the

r-ASRM and the location of the pain for patients diagnosed with endometriosis?

6. PLOS authors have the option to publish the peer review history of their article (what does this mean?). If published, this will include your full peer review and any attached files.

Reviewer #1: **Yes: **Samaneh Rokhgireh

Reviewer #2: No

Reviewer #3: **Yes: **Roya Derakhshan

---

## [Author Response · Author response to Decision Letter 0]

11 Jan 2024

December 28, 2023

Federico Romano, M.D., Ph.D.

Academic Editor

PLOS ONE

Dear Dr. Romano:

Thank you for soliciting the reviews. The comments have helped us better frame our work and as a result we believe we have a much stronger manuscript. Therefore, on behalf of my co-authors, I am re-submitting the enclosed article titled “An artificial intelligence approach for investigating multifactorial pain-related features of endometriosis” for possible publication in PLOS ONE. We are encouraged by the Reviewers’ positive responses regarding our manuscript and appreciate the Reviewers' careful critique. We have taken each comment under consideration and have responded directly to each below. We have highlighted all changes to the manuscript to assist in the review process.

The work in this manuscript was previously submitted to PLOS Digital Health. It has not been published elsewhere. All authors contributed to the design of the study, analysis and interpretation of the data, and revision and final approval of the article. All authors attest to the validity and legitimacy of the data and agree to its submission to PLOS ONE.

We thank you for your consideration of our manuscript and look forward to hearing from you. Please let us know if you have any questions.

Sincerely,

Amber Kiser, PhD Candidate

Department of Biomedical Informatics, University of Utah

Karen Eilbeck, PhD, FACMI

Professor

Department of Biomedical Informatics, University of Utah

Journal Requirements:

RESPONSE: Thank you for including these templates. We have re-formatted the manuscript per the style requirements.

Did you know that depositing data in a repository is associated with up to a 25% citation advantage (https://doi.org/10.1371/journal.pone.0230416)? If you’ve not already done so, consider depositing your raw data in a repository to ensure your work is read, appreciated and cited by the largest possible audience. You’ll also earn an Accessible Data icon on your published paper if you deposit your data in any participating repository (https://plos.org/open-science/open-data/#accessible-data).

RESPONSE: We appreciate the suggestion and agree data repositories are a wonderful way to share data and increase reproducibility. The raw ENDO data contains protected health information (PHI) and access is therefore manage by the National Institutes of Health (NIH). Data requests for the ENDO Study can be directed to NICHD/DIPHR Biospecimen Repository Access and Data Sharing (BRADS). The public GitHub repository amberkiser/endometriosis-pain-net contains the underlying code. The Bayesian network can be accessed via the web address: amber-kiser.shinyapps.io/ENDO-pain-app.

RESPONSE: We are grateful for this reminder and have reviewed the funding information to ensure the grant numbers are correct.

"The authors would like to thank Evan Christensen for his assistance in proofreading and editing the manuscript. We want to acknowledge support from the National Library of Medicine training grant (T15LM007124). We thank the Pedigree and Population Resource of Huntsman Cancer Institute, University of Utah (funded in part by the Huntsman Cancer Foundation) for its role in the ongoing collection, maintenance and support of the Utah Population Database (UPDB). We also acknowledge partial support for the UPDB through grant P30 CA2014 from the National Cancer Institute, University of Utah and from the University of Utah’s program in Personalized Health and Center for Clinical and Translational Science."

"ACK is supported by training grant T15LM007124 from the National Library of Medicine. KCS is supported in part by the National Institute on Aging of the National Institutes of Health under Award Number K01AG058781. The ENDO (Endometriosis, Natural History, Diagnosis, and Outcomes) study was funded by the Intramural Research Program, Eunice Kennedy Shriver National Institute of Child Health and Human Development (NICHD), National Institutes of Health (contract numbers N01-DK-6-3428, N01-DK-6-3427, and 10001406-02). The funders had no role in study design, data collection and analysis, decision to publish, or preparation of the manuscript."

RESPONSE: We have corrected the Acknowledgements section of the manuscript and removed all funding-related text. We ask that the Funding Statement remain as it is:

"ACK is supported by training grant T15LM007124 from the National Library of Medicine. KCS is supported in part by the National Institute on Aging of the National Institutes of Health under Award Number K01AG058781. The ENDO (Endometriosis, Natural History, Diagnosis, and Outcomes) study was funded by the Intramural Research Program, Eunice Kennedy Shriver National Institute of Child Health and Human Development (NICHD), National Institutes of Health (contract numbers N01-DK-6-3428, N01-DK-6-3427, and 10001406-02). The funders had no role in study design, data collection and analysis, decision to publish, or preparation of the manuscript."

RESPONSE: We are grateful for the reminder and have reviewed all references to ensure they are correct and complete.

Reviewer Comments - Review Comments to the Author:

Reviewer #1: Dear Author

1-As we know some endometriosis patients do not complain pain and just have fertility problems, how can we use this approach for diagnosis ?

RESPONSE: We thank the reviewer for their question and agree some women with endometriosis only present with fertility problems. This scenario is captured in the network as we included a node for subfertility. In this scenario, only subfertility would be marked as present, returning a relative risk of 1.470 [95% CI 1.389, 1.551] for endometriosis. 

2-In figure 2 some clusters such as number 1,7,12,...were not included.

RESPONSE: We apologize for the confusion. These clusters were removed from the network as less than 25 participants indicated they felt pain regularly in these grouped pain locations. We address this in the Materials and methods section, under Clustering pain map sites, “To reduce extraneous variables, grouped anatomical locations where fewer than 5% of participants (N<25) indicated they felt pain were removed from further analysis.” 

3-why do you consider region neck,breast,arm,sternum as pain region?These area are not site of endometriosis pain

RESPONSE: We are grateful for the reviewer’s comment and question regarding pain regions. The anatomical pain map used for evaluation of pain regions was part of an international standard, as stated in the Materials and methods section, under Pain evaluation, “The VAS and anatomical map were adapted from the Pelvic Pain Assessment Form created by the International Pelvic Pain Society.” This standard evaluation form can be accessed here: https://endometriosis.ucsf.edu/sites/g/files/tkssra1076/f/wysiwyg/IPPS%20assessment%20form_fillable.pdf.

Additionally, while these are not regions commonly where endometrial lesions are found, there are reports of extrapelvic endometriosis (Mangal R, Taskin O, Nezhat C, Franklin R. Laparoscopic vaporization of diaphragmatic endometriosis in a woman with epigastric pain: a case report. Available from: https://www.ncbi.nlm.nih.gov/pubmed/8855079; Soares T, Oliveira MA, Panisset K, Habib N, Rahman S, Klebanoff JS, et al. Diaphragmatic endometriosis and thoracic endometriosis syndrome: a review on diagnosis and treatment. Available from: http://dx.doi.org/10.1515/hmbci-2020-0066) as well as cases of referred pain. We wanted to include all areas in the initial data gathering and clustering to be sure not to miss any important signals.

4-What do you think about including duration of pain,response to previous hormonal treatment,family history of endometriosis to your approach?

RESPONSE: We appreciate the reviewer’s suggestion. These variables could certainly be investigated in future work and added as nodes in a future network. We addressed this in the last paragraph of the Discussion section, “Pain duration as well as pain description (i.e., sharp, dull, radiating) are potentially important variables to include in future studies along with family history of endometriosis and response to previous hormonal treatment.”

Reviewer #2: Dear Editor,

Thanks for selecting me as a manuscript referee entitled " An Artificial Intelligence Approach for Investigating Multifactorial Pain-Related Features of Endometriosis". according to the following comments, the manuscript can be re -reviewed on the condition of modifying the following.

1. The numbers in the tables should be specified, for example which RR number is.

2. Explain the tables to make what they predict.

RESPONSE: We appreciate the reviewer’s comments regarding the tables. We have updated the captions of the tables to clarify what is demonstrated in the table and what they predict. Table 1 caption now reads, “This table demonstrates univariable analysis with the Bayesian network. The numbers represent mean estimates of the relative risk (RR) of the specified diagnosis, i.e., endometriosis, uterine fibroids, benign ovarian cysts, other gynecological pathology, or a normal pelvis, given the presence of a single symptom with 95% confidence intervals shown in parentheses. Darker red denotes increased RR, while the darker blue denotes decreased RR. For the definition of RR, see S4 Fig.” Table 2 caption now reads, “This table demonstrates multivariable analysis with the Bayesian network. The numbers represent mean estimates of the relative risk (RR) of the specified diagnosis, i.e., endometriosis, uterine fibroids, benign ovarian cysts, other gynecological pathology, or a normal pelvis, given a combination of symptoms (presence or absence of a single or multiple symptoms) with 95% confidence intervals shown in parentheses. Darker red denotes increased RR, while the darker blue denotes decreased RR. For the definition of RR, see S5 Fig.”

3. In the whole article, there is a lot of disruption and the clinical application of artificial intelligence in the treatment and diagnosis of endometriosis is not clear.

RESPONSE: We apologize for the confusion and appreciate the reviewer’s feedback. We have updated language in the paper for clarification. The objective of this study was to investigate pain-related symptoms of endometriosis using an artificial intelligence approach, which was a Bayesian network, as stated in the Materials and methods section, under Bayesian network, and in the Conclusions section. While we suggest this Bayesian network could assist in the diagnosis of endometriosis, it is currently a prototype, requiring much more external validation prior to actual clinical application. We clarified this in the Conclusions section stating, “We built a prototype web application to demonstrate the potential of the Bayesian network to assist in diagnosis after external validation.” With faster diagnosis, potentially expedited by non-surgical diagnostic aids like our Bayesian network, more effective treatments of endometriosis can be applied as suggested by a recent qualitative study. (Drabble SJ, Long J, Alele B, O’Cathain A. Constellations of pain: a qualitative study of the complexity of women’s endometriosis-related pain. Br J Pain [Internet]. 2021 Aug;15(3):345–56. Available from: http://dx.doi.org/10.1177/2049463720961413) 

Reviewer #3: In Data Collection:

1-In the data collection, it was said that the inclusion criteria for the study were patients who underwent laparotomy or laparoscopy for example TL, myomectomy etc., Was the pain questionnaire provided to the patients "before" all of the surgery?

RESPONSE: We are grateful for the reviewer’s comment. Yes, the pain evaluations were provided to patients prior to any surgery. As stated in the Materials and methods section, under Pain evaluation, “Prior to surgery, participants answered in-depth survey questions regarding their pain experiences…Additionally, prior to surgery, participants indicated on a computerized anatomical map where they felt pain regularly.”

2-Why was it necessary to exclude patients with endometriosis from the study?

RESPONSE: We appreciate the reviewer’s question. One of the primary aims of the original ENDO study was to assess the relationship between environmental, lifestyle, reproductive (including pelvic pain), and psycho-social factors and endometriosis. In order to prevent recall bias, women who had a prior diagnosis of endometriosis before their laparoscopy/laparotomy were excluded. We clarified this in the Materials and methods section, under Data collection, “To prevent recall bias, women who had a prior surgical diagnosis of endometriosis were excluded.”

3- Is it possible to perform an analysis based on the relationship between the r-ASRM and the location of the pain for patients diagnosed with endometriosis?

RESPONSE: We appreciated the reviewer’s suggestion. This sensitivity analysis is described iin the last paragraph of the Materials and methods section, under Sensitivity analyses, “We evaluated the ability of the Bayesian network to discriminate between rASRM stages, categorized as early-stage endometriosis (rASRM stages I and II) or late-stage endometriosis (rASRM stages III and IV), as well as endometriosis typology, categorized as superficial or deep infiltrating (DIE) and endometriomas (22). We queried the net for the RR of a postoperative diagnosis as previously described. We used the independent samples t-test to evaluate if the RR between each stage or typology were significantly different. We adjusted the p-values to control for the false discovery rate.” 

The full results for this analysis can be seen in S4 Table (Supplemental Information). In the Results section, under Sensitivity analyses, we state, “Results are presented in S4 and S5 Tables. The RRs for early-stage endometriosis were statistically different from the RRs for late-stage endometriosis in 8 out of 17 cases. The RRs for superficial endometriosis were statistically different from the RR for DIE or endometriomas in 16 out of 17 cases. This demonstrates the potential for discrimination between stages or typology. However, more data from each category would be required to develop a network for this purpose.”

---

## [Editor Report · Decision Letter 1]

17 Jan 2024

An artificial intelligence approach for investigating multifactorial pain-related features of endometriosis

PONE-D-23-25577R1

Dear Dr. Eilbeck

We’re pleased to inform you that your manuscript has been judged scientifically suitable for publication and will be formally accepted for publication once it meets all outstanding technical requirements.

Kind regards,

Federico Romano, M.D., Ph.D.

Academic Editor

PLOS ONE

Additional Editor Comments (optional):

Dear Dr. Karen Eilbeck, thank you for considering our suggestions and partially modifying the manuscript. I believe it is now of interest to the scientific community.

---

## [Editor Report · Acceptance letter]

28 Jan 2024

PONE-D-23-25577R1 

PLOS ONE

Dear Dr. Eilbeck, 

I'm pleased to inform you that your manuscript has been deemed suitable for publication in PLOS ONE. Congratulations! Your manuscript is now being handed over to our production team.

Kind regards, 

on behalf of

Dr. Federico Romano 

Academic Editor

PLOS ONE